# Effects of Host-Adaptive Mutations on Hop Stunt Viroid Pathogenicity and Small RNA Biogenesis

**DOI:** 10.3390/ijms21197383

**Published:** 2020-10-06

**Authors:** Zhixiang Zhang, Changjian Xia, Takahiro Matsuda, Akito Taneda, Fumiko Murosaki, Wanying Hou, Robert A. Owens, Shifang Li, Teruo Sano

**Affiliations:** 1State Key Laboratory of Biology of Plant Diseases and Insect Pests, Institute of Plant Protection, Chinese Academy of Agricultural Sciences, Beijing 100193, China; zhangzhixiang02@caas.cn (Z.Z.); jeffreyshya@163.com (C.X.); houwanying@caas.cn (W.H.); 2Plant Pathology Laboratory, Faculty of Agriculture and Life Science, Hirosaki University, Bunkyo-cho 3, Hirosaki 036-8561, Japan; t.m.62-0109-yg@docomo.ne.jp (T.M.); fmk6639@gmail.com (F.M.); 3Graduate School of Science and Technology, Hirosaki University, Bunkyo-cho 3, Hirosaki 036-8561, Japan; taneda@eit.hirosaki-u.ac.jp; 4Molecular Plant Pathology Laboratory, USDA/ARS, Beltsville, MD 20705, USA; owensj301@hotmail.com; 5Environment and Plant Protection Institute of Chinese Academy of Tropical Agricultural Sciences, Haikou 571101, China

**Keywords:** circular RNA, hop, non-coding RNA, viroid, RNA silencing, small RNA

## Abstract

Accidental transmission of hop stunt viroid (HSVd) from grapevine to hop has led to several epidemics of hop stunt disease with convergent evolution of HSVd-g(rape) into HSVd-h(op) containing five mutations. However, the biological function of these five mutations remains unknown. In this study, we compare the biological property of HSVd-g and HSVd-h by bioassay and analyze HSVd-specific small RNA (HSVd-sRNA) using high-throughput sequencing. The bioassay indicated an association of these five mutations with differences in infectivity, replication capacity, and pathogenicity between HSVd-g and HSVd-h, e.g., HSVd-g induced more severe symptoms than HSVd-h in cucumber. Site-directed mutagenesis of HSVd-g showed that the mutation at position 54 increased pathogenicity. HSVd-sRNA analysis of cucumber and hop plants infected with different HSVd variants showed that several sRNA species containing adaptive nucleotides were specifically down-regulated in plants infected with HSVd-h. Several HSVd-sRNAs containing adaptive mutations were predicted to target cucumber genes, but changes in the levels of these genes were not directly correlated with changes in symptom expression. Furthermore, expression levels of two other cucumber genes targeted by HSVd-RNAs, encoding ethylene-responsive transcription factor ERF011, and trihelix transcription factor GTL2, were altered by HSVd infection. The possible relationship between these two genes to HSVd pathogenicity is discussed.

## 1. Introduction

Viroids are a unique group of non-encapsulated and non-coding RNAs whose unusual physical and biological properties have long attracted the interest of plant pathologists and molecular virologists. They are small, single-stranded, circular RNA molecules that are highly intra-molecular base-paired, folding into either a rod-like (family *Pospiviroidae*, replicates in the nucleus) or multi-branched (family *Avsunviroidae*, replicates in chloroplasts) secondary structure with a series of short, double-stranded helices punctuated by loops/bulges containing unpaired nucleotides [1] that provide potential sites for interaction with host proteins [2,3].

Many viroid isolates contain a complex mixture of sequence variants. These sequence variants are formed through point mutation, sequence duplication, and recombination [4], and within a given host, the polymorphic population of viroid sequence variants can be described by the quasispecies model of molecular evolution [5]. How such a polymorphic population is generated in many higher plant species is only partially understood.

Mutation and selection are essential for viroid evolution. Chloroplastic viroids have extremely high mutation rates [6,7], and the mutations rates of nuclear viroids are similar to those of RNA viruses [6]. High mutation rates during replication provide genetic resources for selection, and host selection plays an important role in shaping the structure of the viroid population [5]. Sequential passage through a genetically diverse range of hosts results in the host and tissue selection of distinct sequence variants from the heterogeneous population of citrus exocortis viroid (CEVd) variants in a given isolate [8]. Several subsequent studies with this viroid [9,10,11], as well as citrus dwarfing viroid (CDVd) [12], also confirmed that viroid populations are host-dependent. For hop stunt viroid (HSVd), although sequence variants from different hosts clustered into several host-specific phylogenetic groups [13,14], direct evidence for the role of host selection in forming HSVd population is lacking.

Although the mechanism of viroid pathogenesis has remained elusive, several models have been proposed to explain this phenomenon. Lack of mRNA activity implies that disease symptoms are induced by the direct interaction of sequence/structure determinant(s) in the viroid genome with currently-unidentified host factors. Pathogenicity determinant(s) possibly involved in such an interaction has been identified for several viroids in both the *Pospiviroidae* and *Avsunviroidae* families [15,16,17,18]. However, the relationship between viroid sequence/structure and pathogenicity are likely complex because multiple structural domains, rather than a single domain, are involved in the regulation of viroid pathogenicity [19]. In the case of citrus cachexia disease, the pathogenicity determinant of HSVd includes five to six nucleotides in the variable domain that differentiate pathogenic (CVd-IIb) from non-pathogenic (CVd-IIa) HSVd variants [20,21]. Further investigation by site-directed mutagenesis showed that symptom severity was affected by subtle changes within this motif, especially the single U to C nucleotide change at position 197 in a ‘cachexia’ HSVd variant M0 [22].

RNA silencing has also been proposed to mediate viroid pathogenicity [23,24,25,26]. Indeed, the involvement of RNA silencing in the induction of viroid symptoms has been experimentally verified for one member of the family *Avsunviroidae*. Small RNA (sRNA) species containing the pathogenic determinant of peach latent mosaic viroid (PLMVd) have been shown to target mRNA encoding the chloroplastic heat-shock protein 90 (cHSP90) for cleavage [27]. More recently, Delgado, et al. [28] identified a similar mechanism for PLMVd-induced peach yellow mosaic (PYM); in this case, cleavage targeted the mRNA encoding a thylakoid translocase subunit required for chloroplast development. Several studies have also implicated RNA silencing in pathogenicity of members of the family *Pospiviroidae* [29,30,31], but evidence for its involvement is less clear than for members of the family *Avsunviroidae* [26,28]. In this regard, the recent demonstration that silencing of transcription factor StTCP23 expression by several sRNA species derived from the virulence modulating region of potato spindle tuber viroid (PSTVd) is associated with symptom development in potato [32] provides solid support.

Phylogenetic analysis of various HSVd isolates endemic in hops found these isolates to be closely-related to HSVd-g(rapevine), implying that hop stunt disease originated by chance transfer of the grapevine isolate into hop [33]. Analysis of changes in the sequence of HSVd-g during 15 years of persistent infection in hop revealed the process by which the initial variant introduced into hop evolved into HSVd-h(op), the variant currently epidemic in commercial hop varieties in the world [34]. Five specific sequence differences between HSVd-g and HSVd-h were identified, but their effect(s) on the biological functions of HSVd remained to be fully understood. Here, we compared the biological property of HSVd-g and HSVd-h by bioassays in cucumbers (indicator host of HSVd) and hop and analyzed HSVd-specific sRNA (HSVd-sRNA) accumulating in the different infected plants by high-throughput sequencing. The five hop-adaptive mutations were shown to affect both HSVd pathogenicity in cucumbers and the biogenesis of sRNA containing these mutations. Detailed analysis of the resulting viroid-specific sRNA profiles identified two cucumber genes predicted to be targeted by HSVd-sRNAs that may be associated with HSVd pathogenicity.

## 2. Results

### 2.1. Biological Properties of HSVd-g and HSVd-h in Cucumber and Hop

To assess the possible influence of adaptive mutations on HSVd pathogenicity, infectivity, and sequence stability, cucumber and hop plants were inoculated with HSVd-g or HSVd-h (Figure 1A). Figure 1B and Appendix A show the typical symptoms of HSVd infection in cucumbers; e.g., crumpled flowers, reduced leaf size, and stunting. Note that stunting is more severe in plants infected with HSVd-g (Figure 1C). As shown in Figure 1D, the first cucumber seedling inoculated with HSVd-g started to exhibit primary leaf curling at 18 days post-inoculation (dpi), and by 31 dpi all five plants showed obvious stunting and leaf curling. In contrast, symptoms of infection by HSVd-h first appeared at 22 dpi, indicating that HSVd-g is more pathogenic than HSVd-h in cucumbers. Stunting was also more severe in plants infected with HSVd-g (Figure 1B,C).

In hop, none of the plants inoculated with either variant showed obvious stunting at eight weeks post-inoculation (wpi). As hop is a perennial crop, the inoculated stocks were over-wintered under greenhouse conditions to continue observations during two subsequent growing seasons. The severity of stunting (i.e., reduction in bine length) varied from plant to plant and season to season. Using the same groups of plants, we also compared the relative rates of progeny accumulation for HSVd-g and HSVd-h by northern-blot hybridization. At 4 wpi, the levels of HSVd-g and HSVd-h progeny in all inoculated cucumber or hop plants were detectable, but the rates of HSVd accumulation in these two hosts were different (Figure 2A,B). Consistent with the delay in symptom expression, HSVd-h accumulated more slowly than HSVd-g in the infected cucumber plants, but both variants eventually reached similar levels (upper panels of Figure 2C,D). Rates of accumulation for these two variants in hop, in contrast, were almost identical.

We further compared accumulation levels of HSVd-sRNA using pooled tissue samples collected from three hop and two cucumber plants at 5 wpi and 8 wpi, respectively. As shown in lower panels of Figure 2C,D, accumulation of HSVd-sRNA ranging in size from ca. 21–24-nt paralleled that of the genomic RNA in plants infected with either HSVd-g or HSVd-h. Levels of HSVd-sRNA in cucumbers and hop appeared similar. Taken together, these results indicate that the five mutations associated with adaptation to hop adversely affected the rate of HSVd accumulation in cucumbers, but affected neither the rate of accumulation in hop nor final progeny levels in either cucumber or hop plants.

### 2.2. Relative Competitiveness of HSVd-g and HSVd-h

To examine whether the adaptive mutations influence the replicative competitiveness of HSVd, we carried out a series of replication competition assays between HSVd-g and HSVd-h. Cucumber and hop plants were inoculated by mixed infectious RNA transcripts of these two HSVd variants in ratios of 1:5, 1:1, or 5:1. Ten cDNA clones from each plant were randomly selected for sequence analysis. HSVd-g was the predominant sequence in each of the progeny populations isolated from almost all cucumber plants (Figure 3A). A second bioassay produced similar results. In hop, HSVd-g was the predominant sequence (70–90%) in each of the progeny populations isolated from the plants inoculated with a 5:1 mixture of RNA transcripts, but not from the plants inoculated with 1:1 and 1:5 mixtures of RNA transcripts (Figure 3B). Although levels of HSVd-h in progeny populations isolated from hop were generally higher than those isolated from cucumbers, HSVd-h failed to out-compete HSVd-g during the relatively short 2-month incubation period in hop plants even when the inoculum contained five-fold more HSVd-h than HSVd-g. The small number of hop plants available for this bioassay (only four plants/treatment) may explain the discrepancy between the results, shown in the middle and right panels of Figure 3B. In short, HSVd-g has replicative advantages to HSVd-h, especially in cucumbers.

### 2.3. Spontaneous Mutations of HSVd-g and HSVd-h in Hop and Cucumber

Spontaneous mutations generated during viroid replication supply the starting material for host selection, and their characterization can help define the process of host adaptation. Thus, we examined in greater detail the mutations that spontaneously appeared in HSVd-g and HSVd-h during infection in both hop and cucumber plants. In hop, 213 independent cDNA clones (107 from HSVd-g and 106 from HSVd-h, 25–30 per plant) were randomly selected from the eight stocks (four each inoculated with HSVd-g or HSVd-h). All plants contained mutants, most of which were singleton mutations. Overall mutation frequency during the term was about 2.2 × 10^−3^ per site in HSVd-g (i.e., 69 mutations in 31,774 nucleotide positions) and 1.4 × 10^−3^ per site in HSVd-h (i.e., 44 mutations in 31,482 positions), indicating that HSVd-h was maintained more stably than HSVd-g in hop.

As shown in Figure 4A, the highest mutation frequency in HSVd-g was found at position 54; i.e., eight of 107 HSVd-g cDNA clones contained a G to A substitution at this position. In contrast, none of the 106 HSVd-h cDNAs contained the mutation at this position, confirming that this G54A transition was selected by hop [33]. The next highest frequencies were observed at positions 205 and 281 in HSVd-g and 281 in HSVd-h. Interestingly, a comparison of the upper and lower panels in Figure 4A indicates that the U/A or A/U transversions at position 281 were apparently reversible. Note also that, of the five adaptive mutations in HSVd-g (i.e., those at positions 25, 26, 54, 193, and 281), only the mutation at position 26 had not yet appeared during the second growing season following its transfer to hop.

In cucumbers, 103 independent cDNA clones (53 from HSVd-g and 50 from HSVd-h, 7–14 per plant) were also obtained from eight cucumber plants (four each inoculated with HSVd-g or HSVd-h) at 4 wpi. The results of the sequence analysis are summarized in Figure 4B. Overall mutation frequency during this period was about 2.0 × 10^−3^ per site in HSVd-g (i.e., 32 in 15,741) and 1.8 × 10^−3^ per site in HSVd-h (i.e., 27 in 14,850). The highest mutation frequencies in HSVd-g were observed at positions 44 and 281; i.e., nine of 53 cDNA clones contained an additional A residue between positions 39 and 44, and nine others contained a U/A transversion at position 281. For HSVd-h at position 44; i.e., six of 50 clones contained an additional A residue between positions 39 and 44, indicating that this insertion is reversible. One HSVd-g cDNA clone contained a G/A substitution at position 54, but none of the HSVd-h cDNA clones examined contained this change.

When comparing Figure 4A,B, it is important to keep several facts in mind. The overall mutation frequency of HSVd-h in hop is significantly lower than in cucumbers or that of HSVd-g in either host. This difference appears to reflect lower selection pressure on HSVd-h in its adaptive host (hop), even during prolonged infection. Moreover, mutation frequencies were not constant along the genome, but rather biased towards a few positions; for example, positions 54, 205, and 281 of HSVd-g in hop or position 44 of HSVd-g and HSVd-h in cucumbers. In contrast, no mutations were found in the upper strand of CCR (Figure 1A) and adjacent regions (positions 60 to 110 in Figure 4A,B), indicating strong selection pressure against mutations in this region. Obvious differences in the distribution patterns of mutations in HSVd-g and HSVd-h other than those at positions 25, 26, 54, 193, and 281, which distinguish these two variants in both hop and cucumber, indicate that these mutations arise in a spontaneous and random way.

### 2.4. Effect of Each Adaptive Mutation on HSVd Pathogenicity

Differences between HSVd-g and HSVd-h pathogenicity in cucumbers must be somehow associated with the one or more of the five adaptive mutations. Thus, we changed single nucleotides at five positions (25, 26, 54, 193, and 281) in HSVd-g to those in HSVd-h, resulting in five mutants: HSVd-g25, -g26, -g54, -g193, and -g281. These five mutants, together with HSVd-g and HSVd-h, were individually inoculated into cucumber seedlings (five plants per mutant). Four weeks later, RT-PCR detection and northern-blot hybridization revealed that all inoculated plants had become infected, indicating that none of the nucleotide changes abolished the infectivity of HSVd-g in cucumbers. As shown in Figure 5, however, cucumber plants infected with the various HSVd-g mutants expressed different symptoms. Mutants HSVd-g25, -g26, -g193, and -g281 all induced mild symptoms resembling those of HSVd-h. HSVd-g54, in contrast, induced a marked reduction in leaf size and plant stunting, symptoms that were more severe than HSVd-g, although the accumulation level was not significantly different from the others (Appendix A). Differences in the pathogenicity of HSVd-g and HSVd-h in cucumbers are, thus, not the result of any one individual mutation, but rather involve an interaction between two or more mutations.

### 2.5. Characterization of HSVd-sRNA Populations

Given that viroid may induce the disease through RNA silencing [26], the five mutations at positions 25, 26, 54, 193, and 281 may cause the difference in pathogenicity of HSVd-g and HSVd-h, or HSVd-g54 and HSVd-h by altering the biogenesis of HSVd-sRNA containing these five mutations. To compare HSVd-sRNA populations, high-throughput sequencing of sRNA was performed for cucumber (at 4 wpi) and hop (at 8 wpi) plants infected with HSVd-h or HSVd-g (1st sRNA analysis), and for cucumber plants infected with HSVd-h or HSVd-g54 at 14 dpi and 28 dpi (2nd sRNA analysis). Data (SRA accession: PRJNA646051) of these sequencing analyses are summarized in Appendix A. 

In all samples, the total number of reads derived from (+)-strand was higher than that from (−)-strand (Appendix A and Figure 6A, Appendix A). Although each sample yielded a smaller number (~6 million) of clean reads in the 1st sRNA analysis than that (>10 million) in the 2nd sRNA analysis, the ratios of HSVd-sRNA to total sRNA were several times higher in the 1st sRNA analysis than those in the 2nd sRNA analysis. This could be explained by the different strategies of sRNA library construction. Gel-enriched sRNA fraction was used as input materials for the 1st sRNA analysis, while the total RNA for the 2nd sRNA analysis. It should be noted that HSVd-sRNA levels were much higher at 28 dpi than at 14 dpi for both HSVd variants (Appendix A), indicating that HSVd-sRNA levels continued to increase as infection progressed. Consistent with results reported by (Martinez et al., 2010) for HSVd-infected cucumbers (cv. ‘Suyo’), a large majority (>85%) of the HSVd-sRNAs detected were 21- to 24-nt in length. As shown in Figure 6B, Appendix A, re-plotting our data according to read length revealed the most abundant size class of HSVd-sRNA to be 21-nt followed by 22- and 24-nt, indicating the involvement of multiple Dicer-like (DCLs) endonucleases in HSVd-sRNA biogenesis in cucumber and hop plants.

Small RNAs are associated with specific Argonaute (AGO) proteins in vivo, where they guide the resulting AGO complexes to their mRNA targets. In Arabidopsis, the loading of an sRNA into a particular AGO complex is mainly determined by the nature of its 5′-terminal nucleotides (Mi et al., 2008). The bar graphs presented in Figure 6C,D, Appendix A compare the nucleotide compositions at the 5′-termini of HSVd-sRNAs. For HSVd-sRNAs of both (+)- and (−)-polarities, C and U were the predominant 5′-nucleotides except for the 24-nt class where A was slightly more abundant than U.

### 2.6. Distribution of sRNA along the HSVd Genome

To analyze the effects of the five adaptive mutations on the biogenesis of HSVd-sRNA, the 5′-termini of (+)-HSVd-sRNA reads and 3′-termini of (−)-HSVd-sRNA reads both containing 21- to 24-nt were mapped to the corresponding positions on the HSVd genomic and anti-genomic RNAs, and the frequency, expressed as read number per million (RPM), was compared between HSVd-g and HSVd-h in cucumber and hop plants in the 1st sRNA analysis and between HSVd-g54 and HSVd-h in cucumbers in the 2nd sRNA analysis. 

Consistent with the previous observation that the difference of methods used for sRNA library preparation largely determined the bias of sRNA profiling [35], different sRNA profiles of HSVd-h in cucumbers were obtained in the two sRNA analyses (compare Figure 7B and Figure 8B) using different methods to construct sRNA libraries. However, because these biases are systematic and highly reproducible, sRNA profiling is suited for determining relative expression differences between samples [35]. Thus, the data of these two sRNA analyses are shown separate, as shown in Figure 7 and Appendix A for the 1st sRNA analysis, and in Figure 8 for the 2nd sRNA analysis.

In the 1st sRNA analysis, the overall sRNA distribution profiles of HSVd-g and HSVd-h were almost identical in both cucumber and hop (compare Figure 7B and Appendix A), indicating the host-independence of HSVd-sRNA profile. Focusing on the sRNA distributions for HSVd-g and HSVd-h in cucumbers, comparison of the upper panel and lower panel of both genomic and anti-genomic strands (Figure 7B) reveals quite similar patterns. The distribution of HSVd-sRNAs generated from each genome position was strongly biased toward several positions (hotspots) indicated in Figure 7B. e.g., the positions of 126–135 in the genomic strand and the positions of 78, 109, 139 in the anti-genomic strand. Although the overall sRNA distribution profiles for HSVd-g and HSVd-h were quite similar, upon close inspection, it revealed some differences, particularly in those containing the five hop-adaptive mutations at positions 25, 26, 54, 193, and 281 (Figure 7B,C). In the genomic-strand, compared with HSVd-h, HSVd-g produced more amount of sRNA containing nucleotide 193, especially of the 21-nt sRNA starting at the position of 179 and the 22-nt sRNA starting at the position of 193. In the anti-genomic strand, compared with HSVd-h, HSVd-g produced more amount of sRNA containing 25/26, 54, or 281, especially of the 21-nt and 22-nt sRNAs starting at the positions of 51 and 270. Similar results were observed for HSVd-sRNA in hop (Appendix A).

In the 2nd sRNA analysis, for both HSVd-g54 and HSVd-h, sRNA distribution profiles at 14 dpi and 28 dpi were almost identical (Figure 8B), suggesting that HSVd-sRNA populations rapidly reach a dynamic balance between the biogenesis and degradation. Focusing on the sRNA distributions for HSVd-g54 and HSVd-h sRNA at 28 dpi, comparison of the upper panel and lower panel of both genomic and anti-genomic strands (Figure 8B) also reveals quite similar patterns. The distribution of HSVd-sRNAs generated from each genome position was also strongly biased toward several hotspots. Although some of these hotspots were different from those identified in the 1st sRNA analysis, several hotspots were common in the two sRNA analyses, e.g., hotspots at positions 125–135 in genomic strand and those at positions 87–95 in the anti-genomic strand. It should be noted that the sRNA distribution profiles of HSVd-g and HSVd-h in cucumber plants are very similar to those reported for HSVd in citrus limon [36] confirming the host-independence of HSVd-sRNA profile.

Close inspection also revealed several differences, e.g., at positions 191, 260, 262, and 281 (Figure 8B). Of particular interest here, as in the 1st small RNA analysis, was the fact that HSVd-sRNA derived from these differentially expressed positions contain hop-adaptive mutations 193(U/C) and 281(U/A) (Figure 8B,C). Compared with HSVd-h, HSVd-g54 generated more 21-nt sRNAs containing nucleotide at positions 25/26, 193, or 281 in the genomic strand, especially the sRNA starts at the position 262. By contrast, compared with HSVd-h, HSVd-g54 generated less 21- and 22-nt sRNAs containing nucleotide at position 281 in the anti-genomic strand, especially the 21-nt sRNA starting at the position 261 and the 22-nt sRNA starting at the position 260. It should be noted that HSVd-g54 and HSVd-h generated a similar amount of sRNA containing the nucleotide at position 54 regardless of genomic and anti-genomic strands (see the row of 54 in Figure 8C); because the two HSVd variants contain the same nucleotides at this position. Thus, although the two sRNA analyses showed some discrepancies, both results supported that hop-adaptive mutations of HSVd affected the biogenesis of HSVd-sRNA.

### 2.7. Predicted Targets of HSVd-sRNAs

To investigate the possible association of RNA silencing with cucumber symptoms induced by HSVd-infection, we predicted potential target genes of HSVd-sRNA (Appendix A) in the genome of cucumber and compared the expression levels of these genes in cucumbers infected with HSVd-g54 and HSVd-h. Appendix A summarize the results of target predictions for HSVd-g54 and HSVd-h, respectively. Because the predicted targets of 21- and 22-nt sRNAs were almost identical, the comparisons described below use only the 21-nt sRNA data. 

Eighteen sRNA species from HSVd-g54 and 8 sRNA species from HSVd-h were complementary to cucumber gene(s). Although most of these sRNAs contain one or two adaptive mutations (especially the U/C transition at position 193), none of the putative target genes matched sRNAs exhibiting obvious differences in accumulation levels in cucumbers plants infected with these two HSVd variants. Thus, these sRNA species are unlikely to be responsible for differences in pathogenicity between HSVd-h and HSVd-g54. 

Interestingly, several sRNA species-specific for HSVd-g54 and containing the adaptive mutation at position 193, (e.g., those with 5′-terminal positions at 180–184, 187, and 188) were predicted to target four cucumber genes. Indeed, the adaptive mutation at position 193 (U for HSVd-g54 and C for HSVd-h) influenced the complementarity between these sRNAs and their potential target genes, which encode heavy metal-associated isoprenylated plant protein (LOC101219619), glutathione S-transferase (LOC101217848), seed biotin-containing protein (LOC101210050), and protein PLASTID MOVEMENT IMPAIRED 1-RELATED 1(LOC101219438).

A previous study from our group [37] used RNA-seq technology to compare the transcriptomes of cucumber plants infected with HSVd-h or HSVd-g54 at 2 dpi, 14 dpi, and 28 dpi. Using this data, we compared the expression levels of the predicted HSVd-sRNA target genes (Appendix A) in cucumber plants infected with HSVd-h or HSVd-g54. No significant difference in expression (*p* < 0.005) were detected between HSVd-h and HSVd-g54 infected cucumbers for any of the predicted target genes, suggesting that differences in the symptoms observed on cucumbers infected with HSVd-h or HSVd-g54 are unlikely to be associated with genes targeted by HSVd-sRNAs containing adaptive mutations.

Despite these similarities in gene expression between cucumber plants infected with HSVd-h or HSVd-g54, two predicted target genes of 21 nt sRNAs whose 5′-termini are located at positions 107 and 177 (HSVd-sRNA107, and HSVd-sRNA177), one encoding an ethylene-responsive transcription factor ERF011 (LOC101219005) and a second encoding a trihelix transcription factor GTL2 (LOC101211010) (Figure 9), were differentially expressed in HSVd-h and HSVd-g54 infected cucumbers as compared to mock-inoculated cucumbers (Appendix A). The former was up-regulated, and the latter was down-regulated only at 28 dpi.

## 3. Discussion

We have previously presented evidence indicating that several epidemics of hop stunt disease in Japan were caused by the accidental transmission of HSVd isolates from cultivated grapevines to hop [33,34]. During these epidemics, a variant of HSVd-g underwent convergent evolution in hop, resulting in a newly-adaptive mutant named hKFKi (or, more simply in this report, HSVd-h) containing five nucleotide changes. The ‘survival of the fittest’ paradigm of Darwinian evolution in which natural selection favors the best-adapted replicators predicts that HSVd-h should have a higher fitness than HSVd-g in hop; indeed, the original HSVd-g was observed to almost disappear from the population in the infected hop plants [34]. 

Each of these five mutations is located either in or immediately adjacent to an internal loop in the predicted secondary structure of HSVd (Figure 1A), regions of the molecule which previous mutational analyses of PSTVd have shown to contain critical structural motifs featuring non-Watson-Crick base-pairing [38]. To determine whether these adaptive mutations could have conferred upon HSVd-h either a higher specific infectivity or an increased rate of replication/accumulation, we compared the infectivity of HSVd-h and HSVd-g in cucumber and hop plants.

Results of these competition assays only partially matched our predictions, i.e., HSVd-h (the hop-adapted mutant) exhibited a reduced ability to replicate/accumulate in cucumbers (the experimental host). The five adaptive mutations that emerged in hop seemed to have an adverse effect upon the ability of HSVd to replicate/accumulate (or move) in other hosts. This phenomenon may represent a ‘trade-off’ among the several sequence changes that occur during the prolonged infection/adaptation process in hop, a new host for the pathogen. Contrary to our expectations, HSVd-h showed only marginal advantages over HSVd-g in its ability to replicate/accumulate in hop, the adaptive host. Both HSVd variants co-existed almost equally with neither one out-competing the other in replication competition assays carried out in hop, again suggesting that they have very similar abilities to replicate in that host.

The fitness of HSVd-h is similar to (but not greater than) that of the original variant, HSVd-g, in hop. Such a result can be explained by the evolution of viroid populations as quasispecies. For viroids, like RNA viruses [39], most host-selected single-nucleotide mutations are expected to be deleterious or neutral, and only a few are beneficial. Indeed, introducting any one of these five adaptive mutations did not significantly increase the accumulation level of HSVd-g in cucumbers. In addition, host selection should result in a higher-fitness population, not a higher-fitness single sequence variant. Sequence analysis of the progeny of these two HSVd variants in hop and cucumber plants (Figure 4) revealed the presence of many sequence variants, some of which may have higher fitness than the respective parents. Namely, the stability of the mutant-consisting quasispecies may have contributed to the slow transition from HSVd-g to HSVd-h observed in our previous analysis in hop [34].

CVd-Iia and CVd-Iib are two HSVd variants that have been isolated from citrus trees showing cachexia disease [20]. They differ by a single C/U substitution at position 196, and CVd-Iia causes more severe symptoms in *Luffa aegyptiaca* than CVd-Iib. Position 196 in CVd-Iia corresponds to the position of one of the adaptive mutations in HSVd-h (i.e., position 193), and HSVd-g causes more severe symptoms in cucumbers than HSVd-h. Similar to the situation with CVd-Iia and CVd-Iib, the nucleotide at this position in the severe variant (HSVd-g) is a C, while the mild variant (HSVd-h) contains a U. Given that both luffa and cucumbers belong to the family Cucurbitaceae, the nucleotide at position 193 in HSVd-g or position 196 in CVd-Iia appears to be an important pathogenicity determinant, at least in cucurbits. Further support for such a conclusion is provided by the observation that the appearance of cachexia symptoms was correlated with a single nucleotide change of U to C at position 197 in HSVd variant CVd-Iia-117 [22]. The mutations at positions 193 in HSVd-g and 197 in an HSVd variant CVd-Iia-117 occupy essentially equivalent positions. 

This position in the V domain may not be the only determinant of HSVd pathogenicity, however. Characterization of additional citrus HSVd isolates has shown that the exchange of C for U at this position may not be required for HSVd pathogenicity [21]. Also, an A for G exchange at position 54 in HSVd-g, resulting in the mutant HSVd-g54, increased HSVd pathogenicity, indicating that HSVd pathogenicity is not determined by a single nucleotide, but by multiple nucleotides or motifs.

Compared with previous studies in cucumbers [40], grapevine [41], and citrus [36], our characterization of HSVd-sRNA by high-throughput sequencing technology was more comprehensive. Samples were isolated from cucumber infected by two different HSVd variants at two different time points (14 dpi and 28 dpi). As described in previous studies, 21- and 22-nt HSVd-sRNA were the predominant species in all sRNA libraries. 5′-terminal nucleotide analysis of our libraries revealed a clear preference for C (loaded by AGO5), followed by U (AGO1). This result differed from that for citrus, where no clear 5′-terminal nucleotide preference could be identified [36].

The reported distribution patterns of HSVd-sRNA along the HSVd genome vary greatly among different studies. For example, our analyses of samples isolated from cucumber revealed hotspots for (+)-HSVd-sRNA near positions 80, 125–135, and 231, whereas hotspots were located near positions 100 and 200 in those from grapevine [41]. These discrepancies seem to be host-related, but even in cucumbers, different hotspots have been reported [40]. It should be noted, however, that similar distribution patterns of HSVd-sRNA have been reported for different hosts, i.e., cucumbers (this study) and citrus [36]. It is possible that the use of different sequence variants in the various studies may be responsible for these discrepancies. Our observation that the presence/absence of adaptive mutations influences HSVd-sRNA biogenesis is consistent with this possibility. These reported differences in HSVd-sRNA distribution patterns indicate the need for a more careful experimental design of studies of viroid resistance involving RNAi technology.

Although RNA silencing has been experimentally shown to be involved in the pathogenesis of both chloroplastic [27,28] and nuclear [29,30,31,32] viroids, this mechanism does not appear responsible for the differences in pathogenicity of HSVd-h and HSVd-g54. This conclusion is based on the observation that none of the HSVd-sRNA species which differ in sequence between HSVd-g54 and HSVd-h were able to pair with any of the genes differentially expressed between HSVd-g54 and HSVd-h infected cucumber plants (Appendix A). The possibility that RNA silencing may mediate HSVd pathogenesis indirectly deserves further investigation, however.

Symptom expression associated with HSVd infection in *N. benthamiana* has been reported to be dependent on the activity of RNA-dependent RNA polymerase 6 (RDR6) [42], indicating a role for RNA silencing in viroid pathogenicity. Indeed, our study has identified several HSVd-sRNA species that are predicted to target cucumber genes, which were differentially expressed in HSVd-g54 and HSVd-h infected cucumber plants as compared with mock-inoculated controls (Appendix A). Among these, the potential biological roles of HSVd-sRNA107 and HSVd-sRNA177 are particularly noteworthy. They were predicted to target genes encoding ethylene-responsive transcription factor ERF011 (LOC101219005) and trihelix transcription factor GTL2 (LOC101211010). Note that GTL2 is involved in plant response to biotic and abiotic stresses [43,44]. Previous studies have shown that transcription factors may be targets of other vd-sRNA species. For example, PSTVd infection of *N. benthamiana* alters the expression of SANT/HTH Myb mRNA, a plant transcription factor regulating morphogenesis [45]. Also, a potato gene which encodes transcription factor StTCP23 was shown to be targeted by several sRNA species derived from the virulence-modulating region of PSTVd. The silencing of *StTCP23* leads to the development of viroid-like symptoms in potato [32].

In a word, we found that the five adaptive mutations between HSVd-g and HSVd-h have effects on the replication capacity and pathogenicity of HSVd and on the biogenesis of HSVd-sRNA. These results deepen the understanding of the evolution of HSVd; the viroid has the widest host range. Moreover, we confirmed the association of the nucleotide at 193 with HSVd pathogenicity and identified two possible target cucumber genes of two HSVd-sRNA species. These two cucumber genes open a door for identifying the role of RNA silencing in HSVd pathogenicity in the future.

## 4. Materials and Methods

### 4.1. HSVd-g, HSVd-h, and Mutants of HSVd-g

HSVd-g (accession No. AB219944) was originally isolated from cultivated grapevine growing in Japan [33]. The hop-adapted variant, HSVd-h (accession No. AB039271), was later isolated from hop growing in the same country and differed from HSVd-g at five positions; i.e., nucleotides 25, 26, 54, 193, and 281 (Figure 1A) [34]. Full-length HSVd cDNA with *BamH* I termini were incubated with T4 DNA ligase to form tandem dimers, and subsequent cloning into the *BamH* I site of pBluescript II SK(-) produced infectious dimeric cDNA clones pBS-HS-gD (HSVd-g) and pBS-HS-hD (HSVd-h) under the control of T7 RNA polymerase promoter. Plasmid DNA was linearized by digestion with *EcoR* V and transcribed with T7 RNA polymerase according to the manufacturer’s instruction (Invitrogen, Carlsbad, CA, USA) to produce infectious RNA transcripts. RNase-free DNase I (Promega-WI, USA) was added after transcription to remove template DNA, and the RNA transcripts were extracted with phenol:chloroform (1:1, *v*/*v*) and quantified by agarose gel electrophoresis and UV spectrometry before use.

To identify the effects of each of the five mutations that distinguish HSVd-g from HSVd-h on HSVd pathogenicity, five mutants were generated from HSVd-g by site-directed mutagenesis (Fast MultiSite Mutagenesis System, TransGen Biotech Co., Ltd. Beijing, China). Nucleotides at position 25, 26, 54, 193, or 281 in HSVd-g were individually changed into that at the corresponding position in HSVd-h, resulting in five mutants; i.e., HSVd-g25, -g26, -g54, -g193, and -g281.

### 4.2. Plant Materials and Viroid Inoculation

In vitro RNA transcripts were dissolved in 50 mM sodium phosphate buffer (pH 7.0) at a final concentration of 100 ng/µl, and aliquots (20 µl) were used to inoculate individual plants. Young cucumber seedlings were mechanically inoculated on their carborundum (600 mesh)-dusted cotyledons; alternatively, hop cuttings were slash-inoculated on newly growing shoots ca. 10 cm in length. Groups of five cucumbers (cv. ‘Suyo’) seedlings or four hop (cv. ‘Kirin II’) cuttings were inoculated with either HSVd-g or HSVd-h and incubated in a glasshouse (23–35 °C, 16 h day length with supplemental light).

### 4.3. RNA Extraction and Northern Analysis of HSVd Genomic and Viroid-Derived sRNAs

Low molecular weight RNA was extracted from fresh leaf tissue (0.1 g) by homogenization in 2 M LiCl and ethanol precipitation. After denaturation in loading buffer containing 25% (*w*/*v*) urea, the extracted RNA was fractioned by denaturing PAGE in either 7.5% polyacrylamide gels (39:1, acrylamide:bisacrylamide) for genomic RNA or 12% polyacrylamide gels (19:1) for small RNA and then transferred to nylon membranes (Biodyne Plus-Pall), and hybridized with DIG-labeled cRNA probes to detect either genomic or HSVd-specific sRNA of both polarities [46]. Hybridization was performed at 55 °C for genomic RNA or 50 °C for sRNA. The resulting hybridization signals were visualized using a BioRad Chemidoc-XRS imaging system (1–2 h exposure) and quantified using the Quantity One (version 4.6.2) software package.

### 4.4. RT-PCR and Determination of HSVd Genomic Sequences

Low molecular weight RNA extracted from HSVd- or mock-inoculated hop and cucumber leaves was used to amplify full-length HSVd cDNA. First-strand cDNA was synthesized using primer HSV-105M (5′-GCTGGATTCTGAGAAGAGTT-3′, complementary to positions 105–83). Second-strand synthesis and DNA amplification were performed using two primers: HSV-78P (5′-AACCCGGGGCAACTCTTCTC-3′, identical to positions 78–95) and HSV-83M (5′-AACCCGGGGCTCCTTTCTCA-3′, complementary to positions 83–66). After fractionation in 7.5% non-denaturing PAGE and visualization with ethidium bromide staining, full-length RT-PCR products were excised, recovered by overnight elution in 0.5 M ammonium acetate-0.1% SDS followed by ethanol precipitation, and cloned into the pGEM-T vector system (Promega, Tokyo, Japan). Nucleotide sequencing was performed either in-house using Licor 4000I (Li-cor) or ABI prism 310 (ABI) DNA sequencers or externally by Macrogen (Seoul, Korea). The resulting HSVd sequences were aligned using Clustal X version 2.0 [47], and the genetic diversity of HSVd progeny in inoculated plants was analyzed. Secondary structures based on the sequence of HSVd-g were calculated using the Mfold algorithm [48].

### 4.5. High-Throughput Sequencing and Bioinformatics Analysis of HSVd-sRNA

In the 1st sRNA analysis, leaf tissue (5.0 g) harvested from four individual cucumbers (4 wpi) or hop (8 wpi) plants infected with HSVd-h or HSVd-g was used to extract Low molecular weight RNA, followed by the sRNA purification using electrophoresis in 12% polyacrylamide gels (19:1) containing 8 M urea. The sRNA species containing ca. 15- to 30-nt were excised, eluted overnight in 0.5 M ammonium acetate–0.1% SDS, precipitated with ethanol, and finally dissolved in 20 μL RNase-free water. Prepared sRNA samples were sent to Hokkaido System Science (Sapporo, Japan) for sRNA sequencing by an Illumina Genetic Analyzer II (Illumina-San Diego, CA, USA) using an index-sequencing strategy.

The 2nd sRNA analysis used a different strategy of sRNA library construction. Total RNA was used as input materials for constructing the library using NEBNext^®^ Multiplex Small RNA Library Prep Set for Illumina^®^ (NEB, Ipswich, MA, USA) following instructions. Library quality was assessed on the Agilent Bioanalyzer 2100 system using DNA high sensitivity chips. The clustering of the index-coded samples was performed on a cBot Cluster Generation System using TruSeq SR Cluster Kit v3-cBot-HS (Illumia) according to instructions. After cluster generation, the library preparations were sequenced on an Illumina Hiseq 2500 sequencing instrument in Novogene Co., Ltd. (Beijing, China), and 50 bp single-end reads were generated.

Raw reads of fastq format were firstly processed through custom perl and python scripts. In this step, clean reads were obtained by removing reads containing ploy-N, with 5′-adapter contaminants, without 3′-adapter or the insert tag, and low-quality reads from raw data. Then, clean reads were chosen and mapped to either the genomic or anti-genomic strand of HSVd using Bowtie 2 [49] by adding an appropriate extension to the 3′-terminus of a linear reference sequence.

### 4.6. Target Prediction and Analysis of HSVd-sRNAs

To search for cucumber transcripts potentially targeted by (+) and (−) 21-nt and 22-nt HSVd-sRNA (Appendix A) containing the five adaptive mutations, we applied the psRNATarget program [50] to the genomic sequence of the cucumber (Chinese Long) V3 [51]. The setting the Expectation Value to <2.0 assured high-quality hybrids. Annotations for the corresponding cucumber genes were recovered from the Cucurbit Genomics database (CuGenDB) (http://cucurbitgenomics.org) using the genome of cucumbers (Chinese Long) V3. Only the predominant 21- and 22-nt HSVd-sRNA species [36] (Figure 6B) were analyzed. A maximum cutoff score = 2 was chosen, considering the following factors: (i) The duplexes between three PLMVd-sRNA species (PC-sRNA8a, PC-sRNA8b, and PYM-sRNA40) [27,28] and two PSTVd-sRNA species (PSTVd-sRNA45, 46) [32], and their verified targets showed scores from 1 to 2; (ii) a score of 2 is capable of predicting the targets of several cucumber miRNA molecules identified in previous studies [52,53,54] and conserved in other species (including miR156, miR159, miR167, miR164, miR172, miR319, and miR858). Next, we compared the expression levels of these genes in cucumbers infected with HSVd-g54 and HSVd-h using previously published transcriptome data [37].

## Figures and Tables

**Figure 1 ijms-21-07383-f001:**
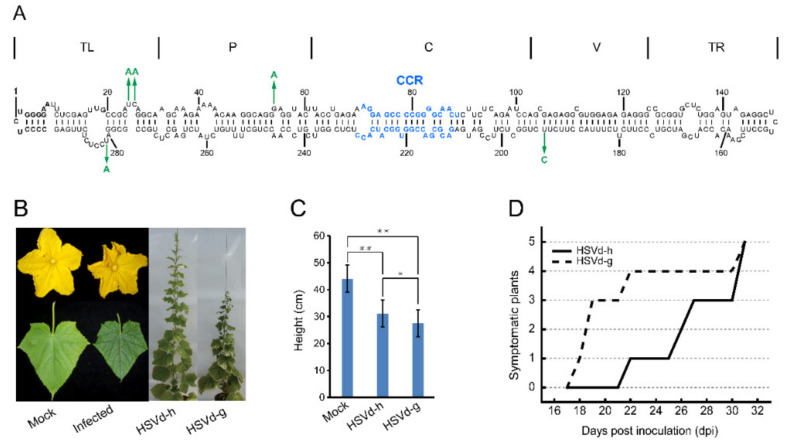
Symptoms of HSVd-g and HSVd-h infection in cucumbers. (**A**) The predicted secondary structure of HSVd-g. The five hop-adaptive mutations between HSVd-g and HSVd-h are indicated by green arrows. From left to right, the five structural/functional domains are terminal left (TL), pathogenicity(P), central (C), variable (V), and terminal right (TR). Central conserved region (CCR) in central domain is indicated by blue color. (**B**) Symptoms on cucumber at 28 days post-inoculation (dpi); left to right, flower, and leaf from healthy plant, flower, and leaf from the infected plant by HSVd-g or HSVd-h, and whole plants infected with HSVd-h or HSVd-g showing shortened internodes at 49 dpi. (**C**) Average height (*n* = 6) of cucumber plants infected with HSVd-h or HSVd-g at 32 dpi. Single or double asterisks indicate statistically significant differences at *p* < 0.05 or 0.01, respectively, in Student’s *t*-test. (**D**) Time course of symptom expression in cucumbers; solid and dashed lines show a number of symptomatic plants inoculated with HSVd-h or HSVd-g, respectively.

**Figure 2 ijms-21-07383-f002:**
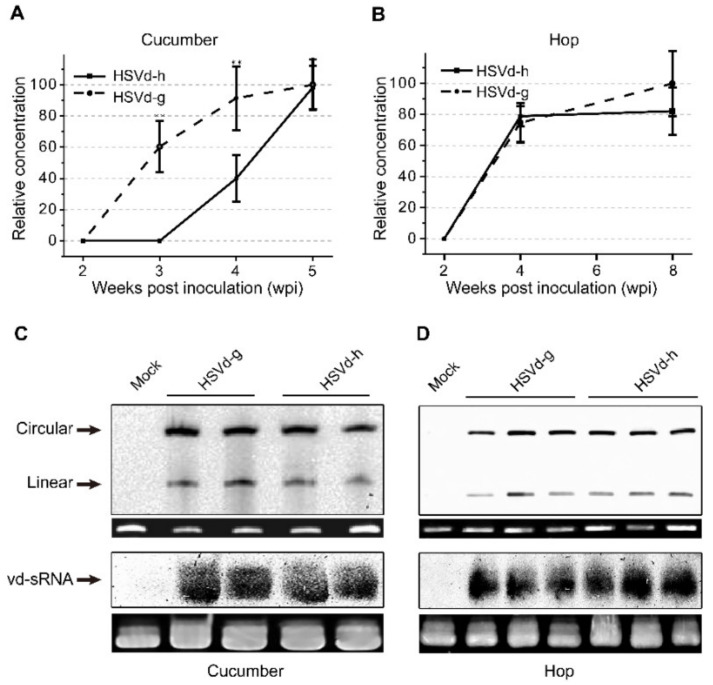
Accumulation of HSVd genomic RNA and viroid-derived small RNA (vd-sRNA) in cucumbers and hop. (**A**,**B**) Accumulation of HSVd genomic RNA in cucumbers (**A**) from 2 to 5 weeks post-inoculation (wpi) and in hop (**B**) from 2 to 8 wpi. Solid line, HSVd-h; dashed line, HSVd-g. Vertical axis indicates relative hybridization signal intensity (pixels/cm^2^) measured by chemiluminescence. Error bars indicate standard deviation. Double asterisks indicate statistically significant differences at *p* < 0.01. (**C**,**D**) Accumulation of HSVd genomic RNA (circular and linear forms) and vd-sRNA in hop at 8 wpi and cucumber at 5 wpi.

**Figure 3 ijms-21-07383-f003:**
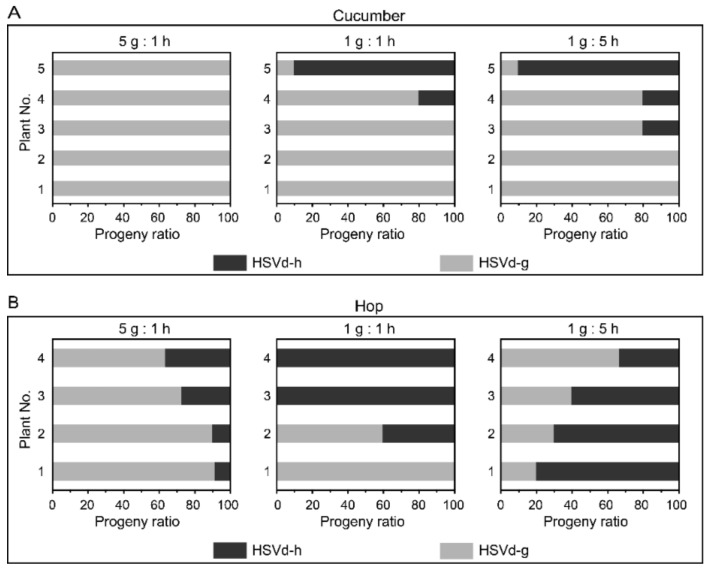
The relative competitiveness of HSVd-g and HSVd-h in different hosts. (**A**) Competition assay in cucumbers. From left to right, five plants were inoculated with mixtures containing HSVd-g and HSVd-h in ratios of 5:1, 1:1, and 1:5. Ten HSVd cDNA clones from each plant were selected at random for sequence analysis, and the horizontal columns indicate the ratio of HSVd-g (light) and HSVd-h (dark) progeny present in individual plants. (**B**) Competition assay in hop. Experimental details as described above, except for the number of plants inoculated (four).

**Figure 4 ijms-21-07383-f004:**
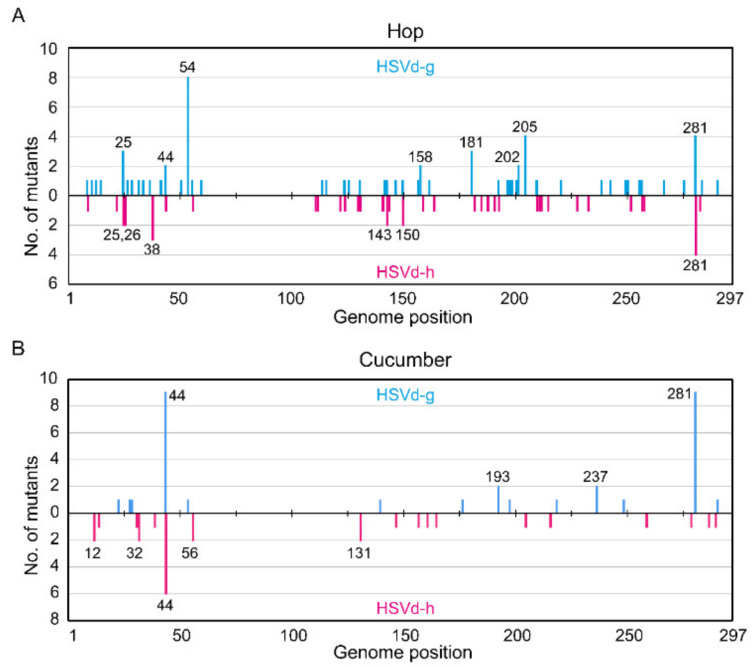
Frequency of mutations in HSVd during the initial stage of replication in (**A**) hop (the second growing season) and (**B**) cucumber (the second passage) plants. The vertical axis indicates the number of cDNA clones containing mutation(s). The numbers of cDNA clones with mutations were plotted along the linearized HSVd genome. Blue bars indicate mutations found in HSVd-g, and red bars indicate those in HSVd-h. The numbers above the bars indicate the nucleotide position of HSVd.

**Figure 5 ijms-21-07383-f005:**
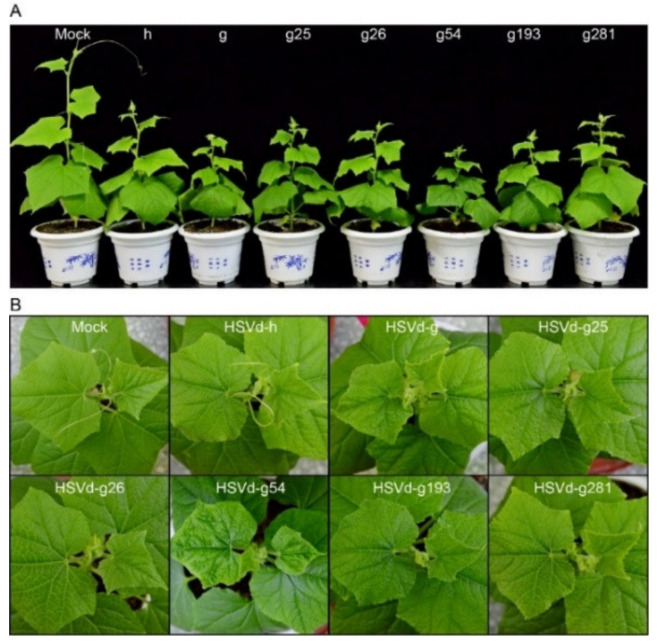
Symptoms on cucumber plants infected with HSVd-h, HSVd-g, or HSVd-g mutants at 28 dpi. HSVd-g mutants HSVd-g25, -g26, -g54, -g193, and -g281 were produced by site-directed mutagenesis. (**A**) Effect of infection on plant height. Abbreviated names of HSVd-h (h), HSVd-g (g), and HSVd-g mutants are shown above. (**B**) Symptoms on uppermost leaves of infected plants.

**Figure 6 ijms-21-07383-f006:**
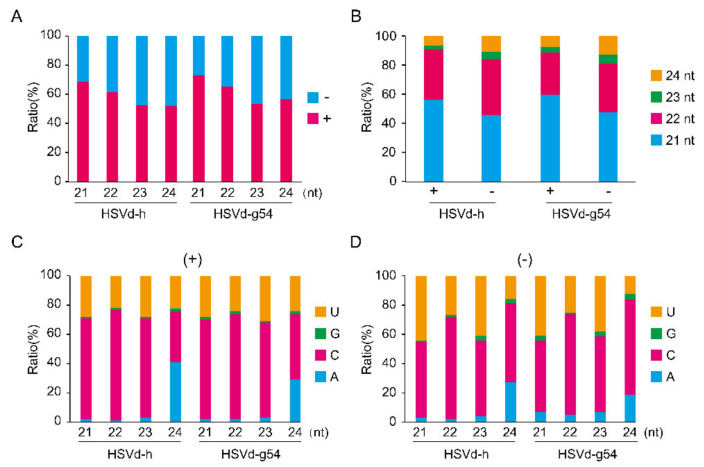
Polarity (**A**), size (**B**), and 5′-terminal nucleotides for genomic (+)-strand (**C**) and anti-genomic (−)-strand (**D**) HSVd-derived small RNA (HSVd-sRNA) extracted from cucumber plants infected with HSVd-g54 or HSVd-h at 28 dpi.

**Figure 7 ijms-21-07383-f007:**
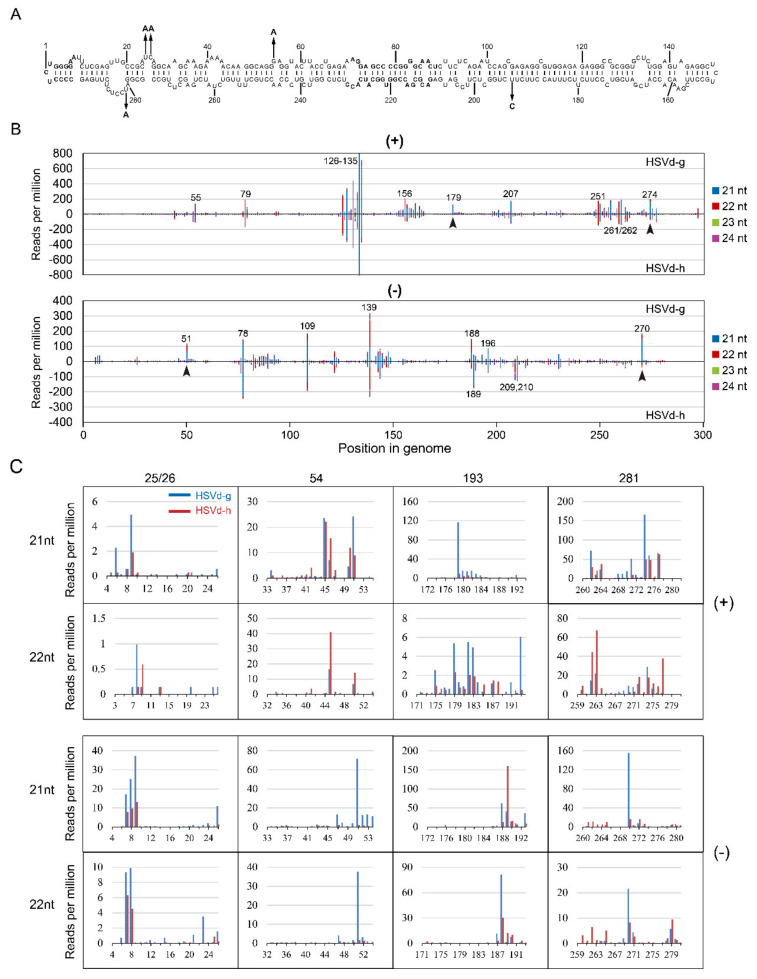
Sequence profiles of HSVd-sRNA derived from the genomic (+) and anti-genomic (−)-strands of HSVd-g or HSVd-h in cucumbers. The top panel (**A**) shows the predicted secondary structure of HSVd-g. The five hop-adaptive mutations between HSVd-g and HSVd-h are indicated by the arrows. sRNA distribution profiles shown in panel (**B**) represent the sum of 21–24-nt HSVd-sRNA populations recovered from cucumber plants infected with HSVd-h or HSVd-g plotted along the linearized HSVd genome (nucleotides 1–297 from left to right). sRNA distribution profiles of HSVd-g and HSVd-h are compared for genomic (upper panel) and anti-genomic (lower panel) strands. Genome positions of selected hot-spot peaks are marked by corresponding numbers. Arrowheads indicate sRNAs containing adaptive mutations that distinguish HSVd-g from HSVd-h and accumulate to obviously different levels. Details of their amounts and sequences are shown in panel (**C**).

**Figure 8 ijms-21-07383-f008:**
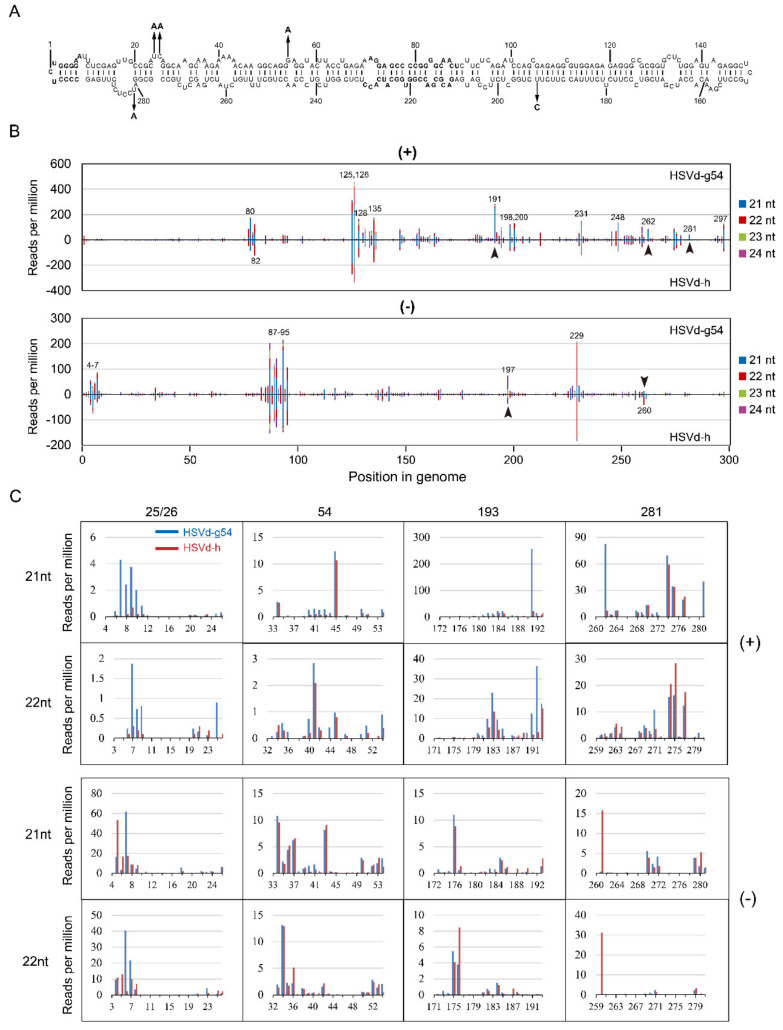
Sequence profiles of HSVd-sRNA derived from the genomic (+) and anti-genomic (−)-strands of HSVd-g54 or HSVd-h in cucumbers at 28 dpi. The top panel (**A**) shows the predicted secondary structure of HSVd-g. The five hop-adaptive mutations between HSVd-g and HSVd-h are indicated by the arrows. sRNA distribution profiles shown in panel (**B**) represent the sum of 21–24-nt HSVd-sRNA populations recovered from cucumber plants infected with HSVd-h or HSVd-g54 plotted along the linearized HSVd genome (nucleotides 1–297 from left to right). sRNA distribution profiles of HSVd-g54 and HSVd-h are compared for genomic (upper panel) and anti-genomic (lower panel) strands. Genome positions of selected hot-spot peaks are marked by corresponding numbers. Arrowheads indicate sRNAs containing adaptive mutations that distinguish HSVd-g54 from HSVd-h and accumulate to obviously different levels. Details of their amounts and sequences are shown in panel (**C**).

**Figure 9 ijms-21-07383-f009:**
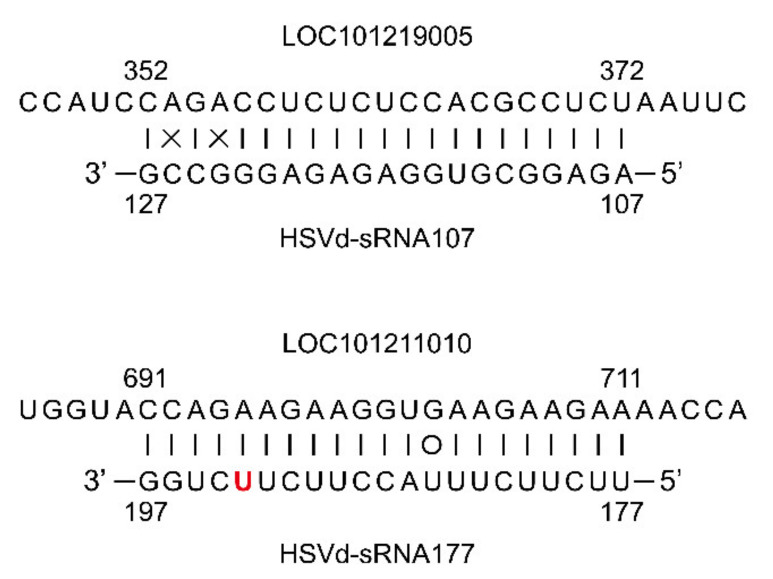
Two predicted target genes of HSVd-sRNAs. Complementary basepairs between HSVd-sRNA107 and HSVd-sRNA177 and their predicted target genes. LOC101219005 encodes an ethylene-responsive transcription factor ERF011; LOC101211010 encodes a trihelix transcription factor GTL2. Targeted host mRNAs are on top (5’–3’ orientation) and sRNAs below (3’–5’orientation); |, Watson-Crick base pairs; o, G:U wobble base pairs; ×, mismatches. Positions of the respective 5′- and 3′-terminal nucleotides are indicated. The nucleotide in red color indicated the adaptive mutation at position 193.

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
