# Peer review of "Effects of Host-Adaptive Mutations on Hop Stunt Viroid Pathogenicity and Small RNA Biogenesis"

_ijms, 2020, doi:10.3390/ijms21197383_

Round 1
Reviewer 1 Report
This is an interesting study and the authors have collected a unique dataset using cutting edge methodology. The paper is generally well written and structured. However, in my opinion, the author needs to address the comment mentioned below in context to the manuscript submitted.
- In Introduction, it would be better to also include the rationale behind choosing cucumber plant in the study. (line 85-95)
- Provide the test of significance for HSVd-g with respect to HSVd-h in figure 1c.
- Line 123-124 provide the correct table. table 1 in not matching with the content written.
- Figure 2, a test of significance missing p-value
- Line 141-146 Discrepancies in the citation Supplementary figure. It should be figS3 and in FigS1 legend casually written, is blot figure why error bar is written in legend. Kindly crosscheck the figure legend and their citation thoroughly throughput the manuscript.
Line 266-269 The reason given seems to be elaborated in context to present study. Whether strategies used for library preparation were different and whether HSVd-sRNA level % is normalized.
Line 307 includes word "it" after the word inspection in line.
Include the concluding remark and future direction of study in viroid field.
Author Response
Point 1: In ‘Introduction’, it would be better to also include the rationale behind choosing cucumber plant in the study. (line 85-95)
Response 1: Thanks a lot for this suggestion. Cucumber is the indicator host of hop stunt viroid (HSVd). It is why we chose this plant in this study. Thus, ‘indicator host of HSVd’ was added following by ‘cucumber’.
Point 2: Provide the test of significance for HSVd-g with respect to HSVd-h in figure 1c.
Response 2: Thanks for the suggestion. We have the test of significance in Fig 1C. To determine the pathogenicity of HSVd-g and HSVd-h, the bioassay of these two variants in cucumber was indeed repeated for five times, twice in Japan and three times in China, in both illumination incubator and greenhouse. The results were the same that HSVd-g caused more severe symptoms than HSVd-h. Because the marked difference in pathogenicity for HSVd-g and HSVd-h, we initially thought it is not required for the statistical analysis.
Point 3: Line 123-124 provide the correct table. table 1 in not matching with the content written.
Response 3: Sorry for the trouble caused by ‘grapevine’. The words ‘wild grapevine’ in the statements in line 92 have be deleted. Indeed, we also performed the bioassays on wild grapevine and found that only HSVd-g replicated stably in wild grapevine but not HSVd-h, which gradually reverted to HSVd-g. However, before the submission, an expert told us that wild grapevine is different with cultivated grapevine and suggested us not showing this result. Thus, the bioassay data about wild grapevine was removed. Thus, the sentence “Data summarized in Table 1 revealed no differences between uninfected wild grapevines and those infected with HSVd-h or HSVd-g.” was deleted in the revised MS. Moreover, table 1 was also removed without influence on understanding of the MS.
Point 4: Figure 2, a test of significance missing p-value
Response 4: Thanks a lot for the suggestion. The missing p-value (<0.01) was added in revised Fig 2A.
Point 5: Line 141-146 Discrepancies in the citation Supplementary figure. It should be figS3 and in FigS1 legend casually written, is blot figure why error bar is written in legend. Kindly crosscheck the figure legend and their citation thoroughly throughput the manuscript.
Response 5: Thanks a lot for this comment and suggesting. We have carefully checked the figure legend and their citation thoroughly throughput the manuscript and revised the mistakes.
Point 6: Line 266-269 The reason given seems to be elaborated in context to present study. Whether strategies used for library preparation were different and whether HSVd-sRNA level % is normalized.
Response 6: At the time of the first analysis, we used to prepare the small RNA fraction by ourselves and send it for high-throughput sequencing analysis. But after all, thanks to the accumulated knowledge and technical improvement, it was found that total RNA is sufficient for analysis. Hence, the total RNA was used for the second analysis. Please understand that the difference between the two is whether the small RNA fraction is enriched or not, and the respective ratios are not affected.
Point 7: Line 307 includes word "it" after the word inspection in line.
Response 7: The word ‘it’ has been added after the word ‘inspection’ in the revised MS.
Point 8: Include the concluding remark and future direction of study in viroid field.
Response 8: Thanks for the suggestion. Such sentences were added in the end of the revised MS.
Reviewer 2 Report
Review report about Zhang and co-workers manuscript entitled „Effect host-adaptive mutations on hop stunt viroid pathogenicity and small RNA biogenesis”.
This paper investigates a very interesting question: how mutations which were stabilized in a special host can influence the developing symptoms. HSVd variants which were evolved in a hop host is only different in 5 nucleotides to the grapevine strain. In this work symptoms of HSVd hop and HSVd grapevine was compared using a hop and cucumber as a host.
I think that the work is very important, the methods are adequate and most up to date. I have problem with the setup of some experiments, the drawn conclusion, and the importance of the presented data.
I think that the first part of the paper, where results about comparison of the HSVd-h and HSVd-g on cucumber and data about hop as a host is good – however the conclusion here is that the h and g strains have a drastic effect only on cucumber and not on hop. In contrast the statements in line92 and in Table 1 there are no data about grapevine. The initial hypothesis of the authors that the h strain has an increased pathogenicity on hop is not demonstrated by the results, and on cucumber we see the opposite – here the g strain has more severe effect. In the presentation of the symptoms on Fig1B I missed the info which strain was used for infection when the symptoms on leaf are presented and how the mock inoculated plants look like comparing the infected ones on the right side of the picture. Also, on panel C, there are no data for the mock, which is a necessity for the comparison. The Northern results about viroid derived small RNAs are very nice why I would include them in the main part, not in the supplementary.
I think that the competition assay is a very good idea – and can give a nice result on hop and cucumber, suggesting the same that pathogenicity of h strain is more severe in cucumber.
Within the mutations which occurred is important to see that position 54 is very mutable, suggesting that somehow this position is not important, possibly it does not affect the important secondary structure, any nucleotide can accommodate it without losing the replicability of the viroid. The same is with the position 281, which can be mutated in both hosts.
In the next part 5 position in HSVdg was mutated to h, which I would abbreviate indication this change: HSVd-g25h. The result on cucumber showed that mutations at 25.26.193 and 281 reverted the symptoms of HSVdg to HSVdh, indicating that they have an important role in symptom determination. When position 54 was mutated (i don’t see increased symptoms), but the symptom stays the same as it was, why I think that this is the only position from the 5 which is not important.
And this is why I think that the further experiments when they have sequenced small RNAs from infection with this strain is so similar to wild type infection. The experimental design here was not the one which can answer the question. The data analysis here also is misleading and I think we don’t have to discuss the origin on the small RNAs in such detail – why if presented I would put this entire section into supplementary.
The last part, discussing possible targets of the viroid derived small RNAs are also important and can add important details to the pathogenesis of the viroid – however we don’t have to forget that the importance of the viroid is on hop and grapevine, why I would suggest a search on grapevine and hop genome for the same transcription factors – could they be targeted in that hosts also. In paragraph between lines381-387 it is said that there was no difference between the expression level of these predicted targets between HSVd-h and HSVD-g54. However, it is not surprising, because these viroids differed only in 4 nucleotides. Here I think the comparison should be make between a mock and the viroid infected plant and include results, possible make some qRT-PCR for the predicted targets to see where they really affected or not.
As a summary I think that this paper contains a huge amount of important results, which should be considered for publication in MDPI IJMS. But for this I would focus on the effect of the viroid on the model plant, cucumber, because for hop the results does not support the difference on the viroid pathogenicity and there is no data for grapevine.
AS in this current form the paper for me seems not complete and needs a lot of rearrangements and new analyses I would reject it, but I would encourage its resubmission after considering my remarks.
Round 2
Reviewer 2 Report
I am happy that the authors addressed my questions and concerns and corrected the manuscript according to them. In this revised form I think that that the manuscript is in a form what can be accepted in MDPI IJMS.